# Repeated proliferative events ameliorate age-associated accumulation of DNA damage in HSPCs

Shubham Haribhau Mehatre[1], Harsh Agrawal[1], Irene Mariam Roy[1], Sarah Schouteden[2], Satish Khurana[1]

Upon aging, hematopoietic stem cells show accumulation of DNA damage that has been causally linked with their functional decline, with debatable role of proliferative events. In this study, we sought to enquire the effect of increased proliferation rate in hematopoietic stem and progenitor cells (HSPCs) on hematopoietic aging. Multiple rounds of blood withdrawals were performed during adult life to maintain a higher proliferation rate in HSPC population in mice. Our experiments showed little effect of increased proliferation on age-associated functional decline in the hematopoietic system. However, we noted a decrease in the double-strand breaks accumulated with age after the serial bleeding regimen. Analysis of scRNA-Seq data from mouse and human HSPCs showed enrichment of DNA damage response pathways. Importantly, we demonstrate that the induction of HSPC proliferation in aged mice was sufficient to activate the DNA damage response in vivo and decrease the load of double-strand breaks. Hence, these results show that repeated blood withdrawals equivalent to clinical blood donation clear DNA damages without impacting the functioning of HSPCs.

## Introduction

It has been well established that the hematopoietic stem cells (HSCs) exhibit functional decline with age. Along with a poor lymphopoietic function, a number of hematopoietic malignancies have been attributed to aging in the hematopoietic system (Downing et al, 2000; Passegue et al, 2003). The frequency of long-term (LT-) repopulating HSCs identified as Thy-1$^{lo}$Sca-1$^{hi}$Lin$^-$Mac-1$^-$CD4$^-$c-kit$^+$ cells increased in the aged mice with a significantly reduced engraftment potential (Morrison et al, 1996). Furthermore, this compensatory increase in the phenotypic HSCs (CD34$^{-/low}$c-kit$^+$Sca-1$^+$Lin$^-$ cells) during aging was associated with a decline in the lymphoid differentiation potential while retaining self-renewal capacity and engraftability (Sudo et al, 2000). Increased proliferation rates, such as in the case of deletion of cell cycle regulators p21 (Cheng et al, 2000b) and p27 (Cheng et al, 2000a), resulted in premature aging like phenotype pointing toward a connection between proliferative events and functional decline. Deletion of Bmi-1 protein that regulates cell cycle progression by controlling the expression of *Ink4* locus also decreased HSC frequency and lymphopoietic activity in adult BM (Park et al, 2003). In fact, initial experiments had suggested that serial transplantations might be more deleterious to the functioning of HSCs than aging (Harrison et al, 1978), indicating deterioration of function because of repetitive proliferative cycles. This decline in HSC function with age and proliferative events has been attributed, at least in part, to compromised DNA damage response resulting in DNA damage accumulation (Rossi et al, 2007). Mice deficient in the expression of ATM that mediates the response to DNA double-strand breaks (DSBs) showed progressive bone marrow (BM) failure because of poorly functioning HSCs (Ito et al, 2004). Similarly, mice with hypomorphic mutation in DNA ligase IV ($Lig^{Y288C}$) (Nijnik et al, 2007) or deletion of Ku80 (Rossi et al, 2007), both crucial components of the NHEJ repair pathway, showed a decrease in the frequency of functional HSCs. Cumulatively, the accumulation of DNA damage appeared to be a major factor for the loss of HSC function and malformations during aging. The exit from quiescence in HSCs was directly linked to DNA damage and functional decline that could be reversed by clearance of reactive oxygen species (Walter et al, 2015). In support of this, an elegant study from Emmanuelle Passegue and colleagues (Flach et al, 2014) showed that cycling aged HSCs show heightened replicative stress and altered DNA repair mechanisms leading to functional decline. However, contrasting results on the effect of proliferation on DNA damage accumulation have been reported, as upon exit from quiescence, aged HSCs exhibited activation of DNA repair mechanisms and clearance of DNA damage (Beerman et al, 2014). Contrarily, myeloablation-induced cell cycle entries repeated over a long period of time in vivo could lead to the accumulation of DSBs and functional decline (Yanai et al, 2025).

Here, we used a long-term serial bleeding regimen to examine the effect of repeated cell cycle entry in hematopoietic stem and progenitor cells (HSPCs) on hematopoietic aging. Our results show little effect of increased proliferation rates during the lifetime on their functional decline with age. Surprisingly, increased

[1]School of Biology, Indian Institute of Science Education and Research Thiruvananthapuram (IISER TVM), Kerala, India   [2]Inter-departmental Stem Cell Institute, KU Leuven, Leuven, Belgium

Correspondence: satishkhurana@iisertvm.ac.in

proliferation was linked to a significantly reduced load of DSBs in aged HSPC population. Analysis of scRNA-Seq data from young and old mouse HSPCs showed that up-regulation of DNA damage response (DDR) pathways was directly correlated with cell cycle activation. Analysis of scRNA-Seq data from human HSPCs presented very similar results. Most importantly, when induced to proliferate in a short-term blood withdrawal experiment, the HSPCs in aged mice showed rapid clearance of DSBs. Overall, we present evidence that proliferative events might not be central to functional decline in aged HSPCs and can clear DSBs in vivo. These results warrant a relook at the proliferation–DNA damage–HSPC function axis hitherto associated with age-associated functional decline in hematopoietic function.

## Results and Discussion

As blood loss is a known inducer of HSC proliferation (Cheshier et al, 2007), we used a serial bleeding-based regimen to test the effect of physiological demand–driven proliferation on hematopoietic aging (Schematic in Fig 1A). After confirming cell cycle entry in HSPCs (Lin⁻c-kit⁺Sca-1⁺ cells) after repeated blood withdrawals (Fig S1A), we compared the blood cell counts in early middle age (12 mo), late middle age (18 mo), and aged (24 mo) mouse groups that underwent serial bleeding (aged donor) with the age-matched (aged control) and young control mice. At early middle age, we did not observe any difference in WBC, lymphocyte, RBC, hematocrit, eosinophil, monocyte, granulocyte, and hemoglobin levels between aged control and aged donor groups (Fig S1B–I). However, a decrease in WBC, monocyte, and granulocyte counts was noticeable in the two groups of aged mice when compared to young controls (Fig S1B, G, and H). Even at late middle age, we did not notice any change in the blood cell counts from aged donor when compared to aged control groups (Fig S1J–Q). Again, when compared to the young control mice, both of the aged mouse groups showed a significant decline in WBC, lymphocyte, hematocrit, monocyte, and hemoglobin levels (Fig S1J, K, M, O, and Q) and a significant increase in eosinophils and granulocytes (Fig S1N and P). At 24 mo of age also, we did not notice any change in the number of monocytes and granulocytes (Fig S1R and S), whereas a decreased number of WBCs were observed (Fig 1B). A decrease in the WBC counts was reflected in lymphocyte numbers (Fig 1C) with a more robust change in aged donor mice than aged controls. We could also note a consistent decline in erythropoietic activity as there was a decrease in RBC numbers (Fig 1D), along with hematocrit (Fig 1E) and hemoglobin levels in aged mouse groups (Fig S1T). Higher eosinophil numbers (Fig 1F) could be linked with pro-inflammatory state that is known to get established with aging in mice and humans (Ferrucci et al, 2005). With age, decreased erythrocyte output has been reported (Florian et al, 2012) along with an increase in the erythromyeloid precursor cells (Rundberg Nilsson et al, 2016). However, only 5–7% of individuals aged >65 yr were found to be mildly anemic (Tettamanti et al, 2010). This age-associated anemic state could also be attributed to the nutritional status. Overall, we found little effect of serial bleeding on blood cell production.

Along with altered blood cell production, the function and composition of the HSPC population change with age (Morrison et al, 1996; Geiger et al, 2013). A robust increase in the HSC population identified as Lin⁻Sca-1⁺c-kit⁺ (LSK) side population (Chambers et al, 2007), CD34⁻LSK cells (Noda et al, 2009), and CD48⁻CD34⁻EPCR⁺CD150⁺LSK cells has been reported (Dykstra et al, 2011) with no change in short-term (ST-) HSCs (CD34⁺Flk2⁻ LSK cells) (Florian et al, 2012). Therefore, we next analyzed whether the serial bleeding regimen had any effect on the age-associated functional changes in HSPC population (Fig S2A). As compared to the young mice, both groups of aged mice showed a robust increase in the frequency of LSK cells (Fig 1G), LT-HSCs (Figs 1H and S2B), MPP2 (Figs 1I and S2C), ST-HSCs (Figs 1J and S2D), and MPP3/4 (Fig S2E and F) populations. Interestingly, no effect of serial bleeding was observed when aged donor mice were compared with aged controls. Previous studies have reported an increase in the expression of CD150 in HSC population (LSKCD34⁻Flt3⁻ cells) from aged mice (Beerman et al, 2010). To examine whether proliferative stress induced by the serial bleeding regimen affected CD150 expression, we analyzed CD150 levels in CD48⁻LSK cells across experimental groups. Our analysis also showed an increase in the expression of CD150 in the HSPC population in the aged group as compared to the young controls. Importantly, we noted a significantly higher CD150 expression in the HSPC population from the aged donor group (Fig S2G). However, our experiments could not link this increased CD150 expression with any functional change in the HSPC population. Reports aimed at the reversal of hematopoietic aging demonstrated a decline in HSC frequency in aged mice. Inhibition of mTOR signaling using rapamycin in aged mice that rejuvenated the old HSCs also decreased the frequency of HSCs without impacting total cellularity (Chen et al, 2009). However, our results did not show any effect of serial bleeding regimen on the size or frequency of HSPC populations, also indicative of no adverse effect of increased proliferative events on hematopoietic populations upon aging. These results are contrasting to the reports implicating HSC proliferative events in functional loss of HSCs coupled with an increase in the number of phenotypic HSCs (Walter et al, 2015).

In the aged hematopoietic system, myeloid bias linked with altered differentiation potential of HSCs (Rossi et al, 2005), or changes in lineage-committed progenitor populations (Pang et al, 2011) have been extensively reported. We performed flow cytometry analysis of lineage-committed cells in the BM and compared the proportion of cells from myeloid and lymphoid lineage in the three groups of mice (Fig S2H). Our experiments also showed a robust increase in the myeloid cell population in the BM of the aged mouse groups (Fig 1K). Concomitantly, we noted a significant decrease in B-cell population (Fig 1L), whereas no change was observed in the T-cell population (Fig 1M). However, these cell populations remained unaffected by the serial bleeding regimen followed (Fig 1K–M). Hence, these experiments also showed no effect of serial bleeding regimen on the change in lineage composition of the hematopoietic system within the BM over aging.

In vivo hematopoietic reconstitution experiments have been extensively used to establish the age-related decline in HSC function (Sudo et al, 2000). Through in vivo engraftment studies on unsorted BM cells (Morrison et al, 1996) and sorted HSC population

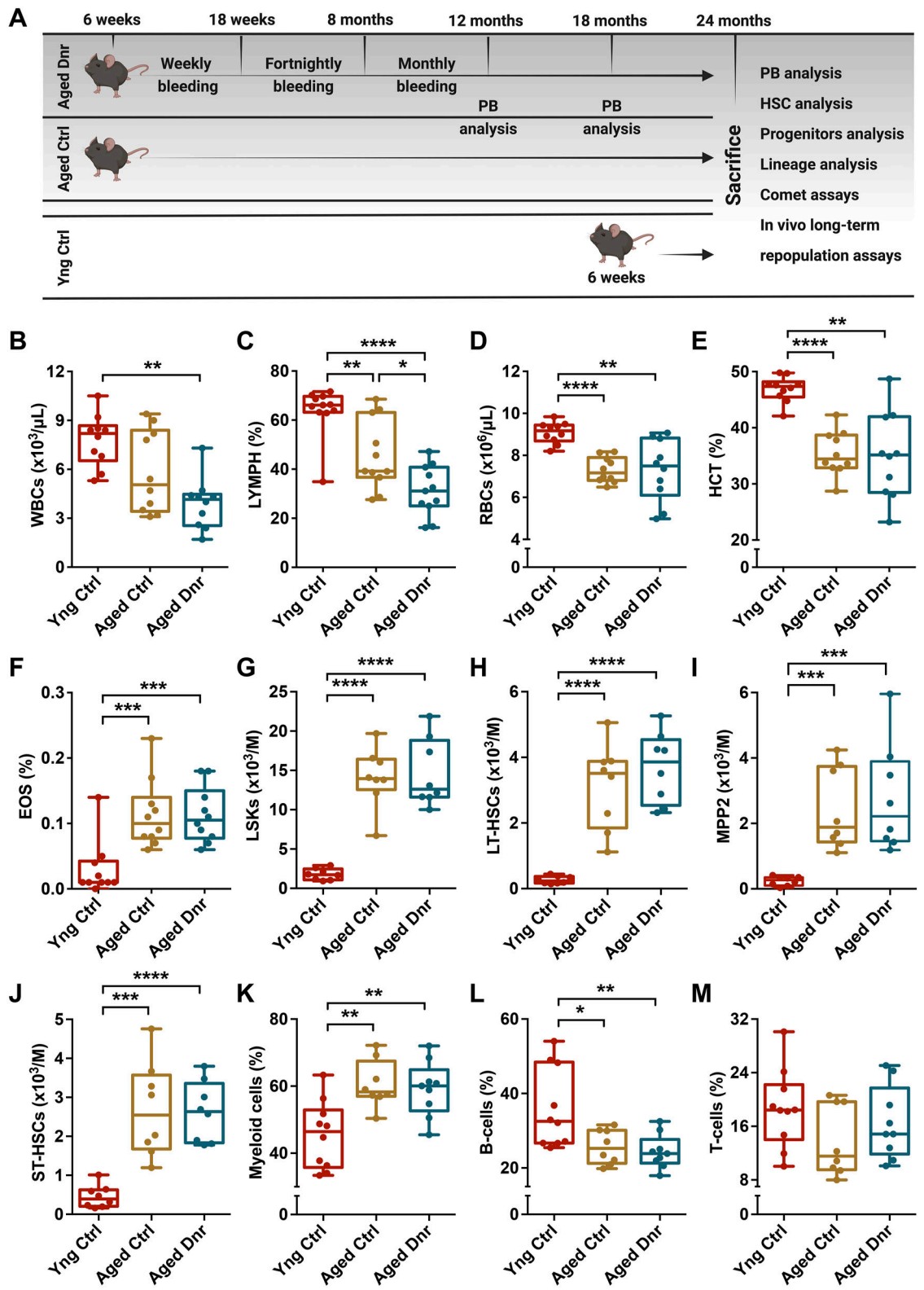

**Figure 1. Serial bleeding does not affect the composition of hematopoietic population over aging.**
**(A)** Schematics depicting the experimental design. Six-week-old mice were subjected to a long-term serial bleeding regimen. To induce excessive rounds of proliferation in hematopoietic stem and progenitor cells (HSPCs) during the lifetime, a long-term serial bleeding regimen was followed. Mice aged 6 wk were used to draw 200 µl blood via the tail clipping method. Blood was drawn weekly for next 12 wk, followed by fortnightly bleeding for 14 wk and monthly bleeding until 1 yr of age. The mice were thereafter maintained up to 2 yr of age, and HSPC analysis was performed at 12, 18, and 24 mo of age. Detailed analysis of the hematopoietic system was followed to assess the effect of enhanced proliferative events on age-associated functional loss in HSPCs. **(B, C, D, E, F)** Aged control and donor groups of mice were

(Dykstra et al, 2011; Ho et al, 2021), decline in aged HSC function has been well documented. We used whole BM MNCs in our repopulation assays to assess whether repeated hematopoietic insult induced by serial bleeding had any adverse impact on aging of HSPC population (Fig 2A). Donor-derived chimerism was analyzed monthly for a period of 4 mo. Up to a period of 8 wk post-transplantation, we did not observe any significant difference in donor-derived engraftment resulting from the transplantation of young versus aged BM MNCs (Fig 2B and C). However, post-transplantation 12 wk onward, we noted a significant decrease in the donor-derived engraftment in both of the aged mouse groups (Fig 2D and E), without any effect of serial bleeding regimen. In addition to a decrease in engraftment, studies have shown a significant change in the lineage reconstitution potential of aged HSCs (de Haan & Van Zant, 1999). A robust decline in B-cell reconstitution without a significant impact on T-cell lineage differentiation has been reported (Poulos et al, 2017). Others have demonstrated a general decline in the lymphoid reconstitution potential (Mann et al, 2018; Ho et al, 2021). This was at least in part attributed to age-associated alterations in the clonal diversity (Muller-Sieburg & Sieburg, 2006). Studies showed the enrichment of myeloid-committed progenitors resulting in myeloid skewing with progressive age (Cho et al, 2008; Gekas & Graf, 2013). We examined multilineage donor-derived engraftment between the three groups of mice after 16 wk of transplantation. We did notice a robust increase in the donor-derived chimerism in myeloid lineage for both the aged mouse groups (Fig 2F). We also noted a concomitant decrease in the reconstitution into lymphoid lineages (Fig 2G and H). These results are consistent with earlier studies that attributed this phenotype to an altered clonal diversity (Muller-Sieburg & Sieburg, 2006) and enrichment of myeloid-committed progenitors (Gekas & Graf, 2013).

Although DNA damage accumulation is reported unequivocally in aged HSCs, a direct link to proliferative events and a causal relation with functional decline in the hematopoietic system have not been investigated. We tested whether the extensive proliferation induced by the serial bleeding regimen altered the status of the age-associated DNA damage accumulation. Most studies on age-associated DNA damage accumulation (Moehrle et al, 2015) used a more sensitive alkaline comet assay with compromise on specificity for DSBs (Olive & Banath, 2006; Lu et al, 2017). We used freshly sorted BM-derived LSK cells from the three groups of mice and performed neutral comet assays to measure DSBs, specifically (Olive et al, 1991) (Fig 3A). We compared the proportion of HSPC population with DNA damage above the threshold (>5% of total DNA in the comet tail), after the long-term bleeding regimen. We noted that the proportion of cells with DSBs was significantly higher in the two aged mouse groups, but unexpectedly decreased after serial bleeding regimen (Fig 3B). In addition, a comparison of DSBs per cell in terms of the Olive tail moment revealed a

significant clearance of DNA damage after serial bleeding regimen (Fig 3C).

As proposed earlier, we attributed this clearance of DSBs to proliferative events during the lifetime (Beerman et al, 2014). To further link DDR pathways with proliferation, we used available single-cell transcriptomic data from aged HSPCs (Hérault et al, 2021). Principal component analysis and gene set enrichment analysis were performed on proliferative and nonproliferative HSPCs identified using the CellCycleScoring function of R-package Seurat (Fig S3A). Analysis showed a robust increase in the overall enrichment of DDR genes (Fig 3D, Table S1) and Hallmark pathways (Fig S3B, Table S2) in proliferative compared with the quiescent HSPCs in aged mice. Interestingly, higher expression of DNA repair genes remained consistent for proliferative HSPCs from young mice as well (Fig 3D). Reactome pathway analysis (Table S3) of up-regulated DDR genes revealed a significant enrichment of DDR pathways in proliferative as compared to the quiescent HSPCs from aged mice (Fig 3E). These results point toward the involvement of proliferation-coupled DDR pathways in clearing the DNA damage accumulated in HSPCs that undergo long dormancy period. We then used scRNA-Seq data from young and aged human HSPCs and compared the gene expression profiles of quiescent and proliferative HSPCs (Fig 3F). This analysis demonstrated extensive similarities between the mouse and human hematopoietic populations, which showed activation of DDR genes in proliferating HSPCs (Fig 3F, Table S4). This was also reflected in the Reactome pathways analysis and showed significant up-regulation of DDR pathways in proliferative compared with the quiescent HSPCs in humans (Fig 3G, Table S5). Our previous results also have described enhanced DDR pathways and DNA repair genes in proliferative fetal liver HSPCs than adult BM HSPCs (Biswas et al, 2020).

Next, we examined whether proliferation induced by short-term serial bleeding regimen in the aged mice could have an impact on DDR gene expression and the status of accumulation of DNA damage. To this end, we first selected some of the DDR genes up-regulated in the proliferative HSPCs in young and aged mice (Fig S3C, Table S1). Notably, most of these genes (*PCNA*, *RPA3*, *RFC4*, *RFC5*, *NEIL3*, *TOPBP1*, *BRCA1*) were also significantly ($P < 0.0001$) up-regulated in the proliferative aged human HSPCs (Fig 3H, Table S4). We performed a short-term serial bleeding regimen in aged mice wherein the mice were bled thrice within a period of 10 d (Fig 4A). We confirmed cell cycle entry of HSPCs after repeated blood withdrawals in aged mice (Fig 4B). We harvested lineage-depleted BM cells to examine the effect of this proliferative regimen on the expression of DDR pathway genes by performing quantitative RT–PCR (Fig 4C–F). Results clearly showed elevated transcript levels of most of the genes across the DDR pathways tested (mismatch repair [MMR]; Fig 4C, base excision repair [BER]; Fig 4D, homologous recombination [HR]; Fig 4E, and nucleotide excision repair [NER]; Fig 4F).

---

compared with young control mice for levels of (B) WBCs, (C) lymphocytes, (D) RBCs, (E) hematocrit, and (F) eosinophils. **(G, H, I, J)** Flow cytometry–based analysis of various HSPC populations in the BM. **(G, H, I, J)** Aged donor mice that underwent serial bleeding regimen were compared with young and aged control mice for the frequency of (G) LSKs (Lin⁻Sca-1⁺c-kit⁺ cells), (H) LT-HSCs (Lin⁻Sca-1⁺c-kit⁺CD150⁺CD48⁻ cells), (I) MPP2 (Lin⁻Sca-1⁺c-kit⁺CD150⁺CD48⁺), and (J) ST-HSCs (Lin⁻Sca-1⁺c-kit⁺CD150⁻CD48⁻). **(K, L, M)** Three groups of mice compared for the proportion of (K) CD11b/Gr-1⁺ myeloid cells, (L) B220⁺ B cells, and (M) CD4/CD8⁺ T cells in the BM. An unpaired two-tailed *t* test was performed. *$P < 0.05$, **$P < 0.01$, ***$P < 0.001$, and ****$P < 0.0001$.

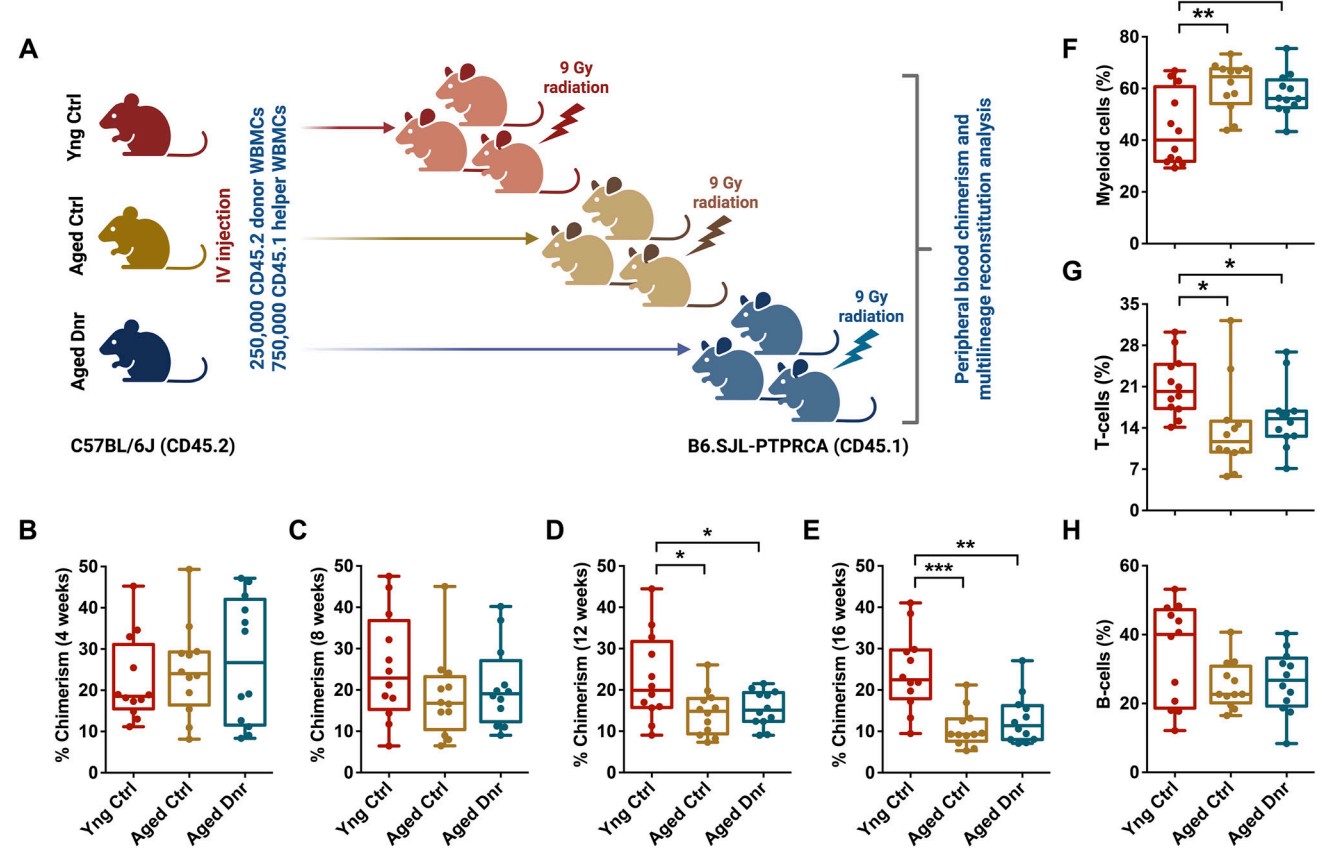

**Figure 2. Repeated proliferative events in hematopoietic stem and progenitor cells during the lifetime do not alter long-term repopulation potential.**
**(A)** Schematic representation of the long-term hematopoietic reconstitution assays performed to compare the function of hematopoietic stem and progenitor cells from aged donor mice with young and aged control mice. Donor-derived (CD45.2) 250,000 whole BM cells were transplanted along with 750,000 (CD45.1) whole BM supporting cells into lethally irradiated (CD45.1) mice. **(B, C, D, E)** Donor-derived peripheral blood chimerism was compared among the three groups of mice after (B) 4 wk, (C) 8 wk, (D) 12 wk, and (E) 16 wk of transplantation. **(F, H)** Comparison of the donor-derived lineage engraftment in the three groups of mice. **(F, G, H)** Flow cytometry analysis was performed to examine the contribution of donor-derived cells within the (F) CD11b/Gr-1$^+$ myeloid cell, (G) CD4/CD8$^+$ T-cell, and (H) B220$^+$ B-cell populations. (n = 8–10). An unpaired two-tailed $t$ test was performed. *$P < 0.05$, **$P < 0.01$, ***$P < 0.001$, and ****$P < 0.0001$.

To further confirm the activation of DNA repair pathways in response to the induction of proliferation in HSPC population, we performed immunostaining to quantify γH2AX foci (Fig 4G). This phosphorylated form of histone variant H2AX indicates the extent of genomic stress and the efficiency of DNA repair pathways, especially in response to DSBs (Mah et al, 2010). We quantified γH2AX foci in the sorted LSKs and assessed whether a short-term bleeding regimen in aged mice activated DNA repair pathway in response to the replicative stress. We observed an increase in γH2AX signals after proliferation, indicating elevated DSB-associated DNA damage response (Fig 4H). It is important to note that an increase in γH2AX foci has also been linked to its ineffective dephosphorylation in aged HSCs, and an indication of altered epigenetic state through a noncanonical function in transcriptional regulation (Flach et al, 2014). When seen together with our results from comet assays that showed clearance of DSBs, we linked increased phosphorylation of H2AX with the activation of DNA repair pathways. To further extend this analysis beyond DSBs, we also performed PARP1 staining, which can reflect activation of the DNA damage response in the context of oxidative DNA damage

response (Hsu et al, 2019) (Fig 4I). Consistent with our γH2AX data, PARP1 staining was also increased upon extensive proliferation, supporting the conclusion that serial bleeding enhances DNA damage response signaling (Fig 4J).

Finally, we examined whether the enhanced DDR pathways could alter the status of DNA damage accumulated in aged HSPCs. We FACS-sorted LSK cells from the mice that underwent short-term bleeding and age-matched controls and performed neutral comet assays (Fig 4K and L). These experiments provided striking results as we noted a massive decrease in the Olive tail moment showing clearance of DSBs (Fig 4K and L). Overall, these results are in contrast to the prevalent notion that higher proliferation can exacerbate DNA damage accumulation. Evidence also shows that the level of DNA damage accumulation might not be directly linked with age-associated functional decline. Contrarily, we show that proliferative events in HSPCs can activate DNA repair mechanisms to reduce the load of DNA damage acquired with age. As mutation accumulation is linked to several hematologic pathologies, serial bleeding as a simple regimen to clear DNA damage can have clinical implications.

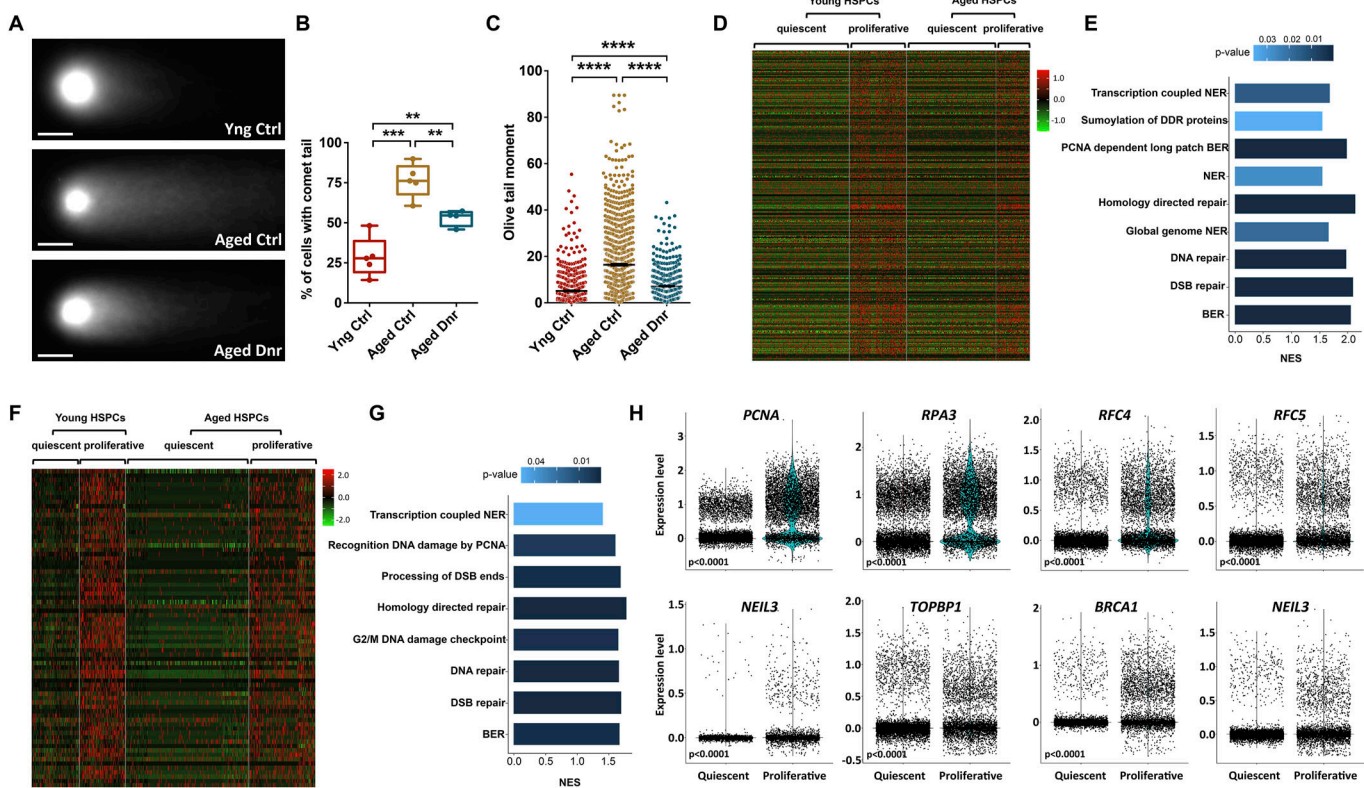

**Figure 3. Proliferative events help in the repair of aging-induced DNA damage.**
**(A, B, C)** Neutral comet assays performed to examine the level of DNA double-strand breaks in hematopoietic stem and progenitor cell (HSPC) population from aged mice that underwent long-term serial bleeding regimen in comparison with young and aged controls. (A) Representative comet images from freshly sorted LSK cells from the BM of mice from each group (scale bar = 20 $\mu$m). (B) Proportion of LSK cells from the three groups of mice that showed comet tail. (C) Comparison of the Olive tail moment as a measure of sheared DNA per cell (n = 4–5, N; young control = 700, aged control = 1,267, and aged donor = 499 LSKs). **(D, E)** Single-cell RNA-Seq analysis to compare the gene expression and pathways enriched in quiescent versus proliferative mouse HSPCs using R-package Seurat. Data on FACS-sorted Flt3⁻Lin⁻Sca-1⁺c-kit⁺ cells from young and aged mice were acquired from public database (Hérault et al, 2021). (D) Heatmap depicting the expression of 409 genes associated with DNA damage response (DDR) in quiescent and proliferative HSPCs from young and aged mice. (E) Differentially expressed Reactome DDR pathways identified using gene set enrichment analysis. The bars depict the normalized enrichment scores (NES) with $P < 0.05$. NES > 0 indicates signature enrichment in proliferative compared with quiescent HSPCs from aged mice. **(F)** Heatmap showing relative expression of 351 DDR pathway genes in quiescent and proliferative HSPCs from young and aged humans. **(G)** Differentially enriched Reactome DDR pathways identified in proliferative compared with quiescent aged human HSPCs using gene set enrichment analysis. The bars depict the normalized enrichment scores (NES) with $P < 0.05$. NES > 0 indicates signature enrichment in proliferative compared with quiescent HSPCs from aged humans. **(H)** Violin plots depicting the feature gene expression of the selected genes associated with DNA damage repair pathways in quiescent and proliferative aged human HSPCs. An unpaired two-tailed $t$ test was performed. $*P < 0.05$, $**P < 0.01$, $***P < 0.001$, and $****P < 0.0001$.

# Materials and Methods

### Mice

Six-week- to 24-mo-old C57BL/6J (CD45.2) and B6.SJL-PTPRCA (CD45.1) mice were bred and maintained in the animal facilities at KU Leuven, Belgium, and IISER Thiruvananthapuram, India. During the experiments, mice were maintained in isolator cages at humidified constant temperature with a 12-h light–dark cycle. The mice were fed with autoclaved water and irradiated food (Safe Diet, France) ad libitum. All animal experiments were approved by the Institutional Animal Ethics Committees for the respective animal facilities. At IISER Thiruvananthapuram, the animals were maintained as per guidelines provided by the Committee for the Purpose of Control and Supervision of Experiments on Animals (CPCSEA), Ministry of Environment and Forests, Government of India.

### Blood withdrawal

To induce excessive rounds of proliferation in HSPCs during the lifetime, a long-term serial bleeding regimen was followed (Fig 1A). Mice aged 6 wk were used for the protocol, and 200 $\mu$l of blood was withdrawn weekly for next 12 wk via the tail clipping method. This was followed by fortnightly bleeding for up to 8 mo of age, and monthly bleeding until the mice reached 12 mo of age. Thereafter, mice were maintained without further intervention (except for blood withdrawal for analysis at 18 mo of age) until euthanasia at 24 mo. Peripheral blood analysis was performed at 12, 18, and 24 mo of age. The interval between the last bleed and bone marrow analysis was 6 mo, as the mice were euthanized at 24 mo and the penultimate PB analysis was performed at 18 mo of age. A short-term serial bleeding regimen of three blood withdrawals in 10-d period with 3-d intervals was used in aged mice (24 mo of age) to

**Figure 4. Short-term proliferative events help in the clearance of aging-induced DNA damage.**
**(A)** Schematic representing short-term blood withdrawal in aged mice. **(B)** Confirmation of the effect of blood withdrawal on cell cycle entry in BM hematopoietic stem and progenitor cells. Aged mice underwent three blood withdrawals within a period of 10 d followed by cell cycle analysis. The mononuclear cells from the BM were harvested, and stained for c-kit, Sca-1, and lineage markers followed by labeling with DAPI and immunostaining for Ki-67. **(C, D, E, F)** Quantitative RT–PCR-based analysis to examine the change in the expression of selected DDR pathway genes in BM-derived lineage-depleted cells after a short-term bleeding regimen in aged mice. **(C, D, E, F)** Expression of genes with function in (C) mismatch repair [MMR] (*Pcna, Rpa3, Rfc4* and *Rfc5*), (D) base excision repair [BER] (*Neil1, Neil3, Xrcc1,* and *Tdg*), (E) homologous recombination [HR] (*Topbp1, Palb2, Mre11a, Brca1,* and *Blm*), and (F) nucleotide excision repair [NER] (*Cetn2, Ddb1*). **(G)** Representative images show LSK cells isolated by FACS from aged control (upper panel) and aged bled (lower panel) mice. The cells were stained with anti-γH2AX antibodies visualizing γH2AX focus formation at double-strand break sites (scale bar = 2 μm). **(H)** Average number of γH2AX-positive foci in LSK cells from aged control and aged bled mice. **(I)** Representative images show LSK cells isolated by FACS from aged control (upper panel) and aged bled (lower panel) mice. The cells were stained with anti-PARP-1 antibodies visualizing DNA damage location and activation in response to DNA damage (scale bar = 2 μm). **(J)** Mean fluorescence intensity (MFI) of PARP-1 in each LSK cells from aged control and aged bled mice was analyzed. **(K, L)** Neutral comet assay performed on FACS-sorted LSK cells from aged mice with or without short-term bleeding regimen followed. (K) Representative comet images showing the level of DNA damage because of double-strand breaks (scale bar = 20 μm). (L) Comparison of the Olive tail moment from neutral comet assays performed on LSK cells from the two groups of mice (n = 3, N; aged control = 964, aged bled = 582 LSKs). An unpaired two-tailed *t* test was performed. *P < 0.05, **P < 0.01, ***P < 0.001, and ****P < 0.0001.

induce proliferation (Fig 4A). At every time point, peripheral blood analysis was performed on Erba hematology analyzer for detailed blood cell counts. Detailed functional analysis of the hematopoietic system was performed at 24 mo of age.

### Bone marrow aspiration

Mice were euthanized via cervical dislocation, and hindlimb bones were harvested. Adjacent muscle tissues were removed, and bones were flushed with 1X PBS using a syringe with 26G needle. The

resulting cell suspension was passed through a 41-μm cell strainer (Corning). The filtered cell suspension was diluted with 1X PBS and centrifuged at 600*g* for 5 min at 4°C. The BM mononuclear cells were carefully resuspended in 1 ml 1X PBS and were counted manually using a Neubauer hemocytometer (Neubauer).

### Transplantation

Frequency and function of the HSPCs from young control, aged control, and aged donor mice were examined using long-term hematopoietic reconstitution assays. Freshly isolated 250,000 whole BM

**Table 1.** List of antibodies used in this study.

| Sr. No. | Antibodies | Source | Catalog number |
|---|---|---|---|
| 1 | BB700-conjugated anti-mouse Sca-1 (Ly6A/E) | BD Pharmingen | 742089 |
| 2 | PECy7-conjugated anti-mouse CD150 | BioLegend | 115914 |
| 3 | PE-conjugated anti-mouse c-kit (CD117) | BioLegend | 105808 |
| 4 | FITC-conjugated anti-mouse CD48 FITC | BioLegend | 103404 |
| 5 | APC-conjugated anti-mouse lineage antibody cocktail | BD Pharmingen | 558074 |
| 6 | APC-conjugated anti-mouse CD11b | BioLegend | 101212 |
| 7 | APC-conjugated anti-mouse Gr-1 (Ly6G) | BioLegend | 108412 |
| 8 | APC-conjugated anti-mouse CD4 | BioLegend | 100516 |
| 9 | APC-conjugated anti-mouse CD8a | BioLegend | 100712 |
| 10 | FITC-conjugated anti-mouse CD45R/B220 | BioLegend | 103206 |
| 11 | AF647-conjugated anti-mouse Ki-67 | BD Pharmingen | 561126 |
| 12 | FITC-conjugated anti-mouse Ter119 | BD Pharmingen | 557915 |
| 13 | APC-conjugated anti-mouse CD45.1 | eBioscience | 17-0453-82 |
| 14 | FITC-conjugated anti-mouse CD45.2 | eBioscience | 11-0454-82 |
| 15 | FITC-conjugated anti-mouse Gr-1/Ly-6G/C | BD Pharmingen | 553126 |
| 16 | FITC-conjugated anti-mouse CD3e | Invitrogen | 11-0031-85 |
| 17 | FITC-conjugated anti-mouse CD41 | BD Pharmingen | 561849 |
| 18 | AF488-conjugated anti-mouse F4/80 | BD Pharmingen | 567201 |
| 19 | Rabbit polyclonal γ-H2AX antibodies | Abcam | ab11174 |
| 20 | Rabbit monoclonal PARP1 antibodies | CST | 46D11 |

Fluorescently conjugated antibodies targeting cell surface markers were employed for the identification of hematopoietic cell populations via flow cytometry.

cells (CD45.2$^+$) were transplanted along with 750,000 competitor WBMCs (CD45.1$^+$) into lethally irradiated (9 Gy) 8- to 12-wk-old female mice. All irradiated mice were fed on enrofloxacin (Baytril) containing water. Peripheral blood chimerism and multilineage engraftment analyses were performed every 4 wk by flow cytometry.

## Flow cytometry

Analysis of donor-derived chimerism, multilineage engraftment, and characterization of hematopoietic system was performed by flow cytometry. For flow cytometry–based analysis, we performed immunostaining on the mononuclear cells followed by fixation of the cells by 2% PFA. In chimerism and multilineage engraftment experiments, donor and recipient cells were identified as CD45.2$^+$ and CD45.1$^+$ cells, respectively. Within the CD45.2$^+$ donor-derived cells, lineage-committed cells were identified as myeloid (CD11b$^+$/Gr-1$^+$), T-cell (CD3e$^+$), and B-cell (B220$^+$) populations. On the basis of the expression of SLAM markers, CD150 and CD48, lin$^-$c-kit$^+$Sca-1$^+$ (LSK) population was subdivided into four subpopulations: CD150$^+$CD48$^-$ (LT-HSCs), CD150$^+$CD48$^+$ (MPP2), CD150$^-$CD48$^+$ (MPP3/4), and CD150$^-$CD48$^-$ (ST-HSCs), which were characterized by allophycocyanin-conjugated anti-lineage antibody cocktail, PE-conjugated anti-mouse c-kit antibodies, BB700-conjugated anti-mouse Sca-1 antibodies, FITC-conjugated anti-mouse CD48 antibodies, and PECy7-conjugated CD150 antibodies (all antibodies used for flow cytometry were made at 1:200 dilution). Suitable isotype controls for each antibody were used in all experiments. A complete list of antibodies used for these experiments is provided in Table 1.

Cell cycle analysis was performed on BM-derived MNC population to examine the effect of serial bleeding regimen on HSPC proliferation. The BM MNCs were first immunolabeled and fixed with 2% PFA for identification of LSK cells, as described above. After cell surface staining with FITC-conjugated anti-lineage antibody cocktail, and BB700-conjugated anti-mouse Sca-1 and PE-conjugated anti-mouse c-kit antibodies, the cells were fixed using BD Cytofix/Cytoperm buffer. The cells were then washed with Perm/Wash buffer and incubated with AF647-conjugated Ki-67 antibodies. The cells were further washed and labeled with DAPI for 30 mins on ice. The samples were analyzed using FACSAria III (BD Biosciences) and FlowJo software (TreeStar).

## Neutral comet assay

This method was used to perform comet assay to assess DNA damage in freshly sorted BM-derived HSPCs under neutral pH. FACS-sorted LSK cells were resuspended in low-melting agarose (type VII; Sigma-Aldrich). The suspension was poured onto agarose-coated comet slides dropwise and incubated at 4°C. After agarose solidifies, slides were immersed in lysis buffer (2.5 M NaCl, 0.1 M EDTA, 10 mM Trizma base, 1% Triton X-100, 10% DMSO, pH 10) for 1 h. The slides are then washed with prechilled neutral electrophoresis buffer and incubated for 30 min. The slides are then transferred for electrophoresis in prechilled 1X

**Table 2. List of primers used in this study.**

| Sr. No. | Primer name | Sequence 5'-3' |
|---|---|---|
| 1 | Mm-Pcna_F | GAGGGTTGGTAGTTGTCGCT |
| | Mm-Pcna_R | CTCAAACATGGTGGCGGAGT |
| 2 | Mm-Rpa3_F | CGCCAGCATGTTACCACAGTA |
| | Mm-Rpa3_R | ATTTCCTCGTCAAGTGGCTCC |
| 3 | Mm-Rfc4_F | GCCAAAGCACAACTGACCAAG |
| | Mm-Rfc4_R | CACTGCAACCACTTCGTCCT |
| 4 | Mm-Rfc5_F | AGAACGCCTTGAGACGAGTG |
| | Mm-Rfc5_R | TCAGAGGGCCAAATCGGAAC |
| 5 | Mm-Neil1_F | AAGGGGCTGGTATTTGGTGG |
| | Mm-Neil1_R | CTCAATGTCAAGCGCAGCTC |
| 6 | Mm-Neil3_F | CGGTGGAAAGCCAACAGAGA |
| | Mm-Neil3_R | ACACATCACACAGCATCCGA |
| 7 | Mm-Xrcc1_F | AAAGAGTGGGTGCTGGACTG |
| | Mm-Xrcc1_R | AGCTTGGGAGCTTCGTCTTC |
| 8 | Mm-Tdg_F | CCCCGATCCTGTGCTATTCTC |
| | Mm-Tdg_R | GTCACGGTTGCCATGTTAGG |
| 9 | Mm-Topbp1_F | AGAGGCTACTGCCCAGAACA |
| | Mm-Topbp1_R | CGAGGCCGTTTGACTACATT |
| 10 | Mm-Palb2_F | GGGAAACGAAAATCAGCCCG |
| | Mm-Palb2_R | AACCACGCCTCTGTTCTGAC |
| 11 | Mm-Mre11a_F | CTGGGAGCGGTTTTCTTGTG |
| | Mm-Mre11a_R | TGGATCTGTGGGGCTCATTT |
| 12 | Mm-Brca1_F | GGCTTGACCCCCAAAGAAGT |
| | Mm-Brca1_R | TGTCCGCTCACACACAAACT |
| 13 | Mm-Blm_F | CTTGGGAGCTGAAAGAGGTG |
| | Mm-Blm_R | AACGAGGAAGAAGCAGTGGA |
| 14 | Mm-Cetn2_F | TGCAGTGGCTTCTTAGTTGTCC |
| | Mm-Cetn2_R | ATGCCACAGCAAGCACTCAT |
| 15 | Mm-Ddb1_F | GTGTCTCAAGAGCCCAAAGC |
| | Mm-Ddb1_R | TCTCTGTGTGGCTGATTTGC |
| 16 | Mm-GAPDH-F | ACCCAGAAGACTGTGGATGG |
| | Mm-GAPDH-R | TTCAGCTCTGGGATGACCTT |

Gene-specific primers were designed to assess the expression of DNA damage repair genes in hematopoietic cell populations from both young and aged mice.

neutral electrophoresis buffer (10X buffer contains 1 M Trizma base, 3 M sodium acetate trihydrate; pH is set at 9 using glacial acetic acid). After electrophoresis (~60 min, 21 V), air-dried and precipitated slides (10X precipitation buffer contains 7.5 M ammonium acetate, which is diluted to 1 M using 100% ethanol) were stained with SYBR Gold solution (1:1,000). The Olive tail moment was scored for 100–200 cells/sample using Open Comet plugin in ImageJ software. In our analysis, we applied a more stringent criterion by classifying cells with >5% tail DNA as damaged.

## Quantitative RT–PCR

Hematopoietic progenitors were isolated with EasySep mouse hematopoietic progenitor isolation kit. Total RNA was prepared using TRIzol reagent. The purity and the concentration of RNA were assessed using a micro-volume spectrophotometer (Colibri, Berthold Technologies GmBH & Co. KG). Two micrograms of RNA from each sample was used to synthesize cDNA using PrimeScript RT reagent kit (Takara Biotechnology Co. Ltd.) according to the manufacturer's protocol. Quantitative PCR was carried out using TB Green Premix (Takara Biotechnology Co. Ltd.). The PCRs were carried out in a CFX96 detection system (Thermal Cycler C1000; Bio-Rad Laboratories). The fold change in gene expression was calculated using the comparative $\Delta C_t$ method. The list of primers used is given in Table 2.

## Immunostaining and imaging

FACS-sorted BM LSK cells were seeded on poly-L-lysine (Sigma-Aldrich)–coated glass slides to promote adhesion, followed by fixation in 2% PFA and permeabilization by Triton X-100 for 10 and 30 min, respectively. The cells were then immunostained using specific antibodies against γH2AX (1:400 dilution), PARP1 (1:400 dilution), and fluorescently labeled secondary antibodies (1:800 dilution). Fluorescence imaging was performed using a Leica STELLARIS 5 DM6 CS upright confocal microscope, and images were captured using Leica oil immersion objective: HCX PL APO CS 63.0x/1.40 Oil CS2 with LAS AF software. A complete list of antibodies used for these experiments is provided in Table 1.

## Image processing and analysis

The 2D images generated by confocal imaging were converted into a .ims file format and analyzed using Imaris (Imaris x64 10.0.0). The 2D images were loaded into the Surpass module, and foci were detected using the "Spots" function with a spot diameter of 0.3 $\mu m$. The total number of γH2AX foci (spot surfaces) was calculated per field of view and normalized to the total number of nuclei to obtain the average foci per cell. For PARP1 analysis, the "Surfaces" tool was applied to generate nuclear masks, and the mean fluorescence intensity (MFI) of each surface was calculated. Quantitative data were exported from Imaris and further analyzed using GraphPad Prism.

## Single-cell RNA-Seq analysis

Single-cell RNA-Seq libraries of BM HSPCs isolated from young and aged mice were retrieved from the Gene Expression Omnibus (GEO; GSE147729) (Hérault et al, 2021) and human HSPC scRNA libraries from the GEO (GSE180298) (Ainciburu et al, 2023). The gene count expression matrix was analyzed using the Seurat v.4.1.1 in RStudio. Cells containing fewer than 100 genes and fewer than 500 unique molecular identifiers were excluded from subsequent analysis. Normalization of raw counts was performed using the NormalizeData function in Seurat. Variable

genes were identified using the FindVariableGenes function. Expression values in the dataset were scaled and centered for dimensional reduction using the ScaleData function in Seurat, with default parameters. Principal component analysis was performed after analysis of cell cycle phase for individual cell was performed with CellCycleScoring function of Seurat, employing a core gene set as previously described (Fan et al, 2018), distinguishing between cells in different stages. As HSPCs at the $G_o$ stage are transcriptionally closer to $G_1$ than S or $G_2/M$ cells, we marked them quiescent, whereas HSPCs in the S and $G_2/M$ phases were identified as proliferative. Differential gene expression analysis was performed with the FindMarkers function. Subsequently, Hallmark and Reactome pathway enrichment analysis was performed with the differential gene expression dataset, employing a fast gene set enrichment analysis package. Pathways demonstrating statistical significance, indicated by a $P ≤ 0.05$, were considered noteworthy.

### Quantification and statistical analysis

Data are represented as the mean ± SEM or as box-and-whisker plots (Min to Max). Comparisons between samples from two groups with normally distributed data with equal variance were made using the unpaired two-tailed *t* test. The Mann–Whitney test was used for comparing two groups where data were non-normally distributed. Statistical analyses were performed with Microsoft Excel, GraphPad Prism 6, and RStudio. For all analyses, *P*-values < 0.05 were accepted as statistically significant.

## Data Availability

The published datasets used for the analysis were retrieved from the GEO using accession numbers GSE147729 (Hérault et al, 2021) and GSE180298 (Ainciburu et al, 2023).

### Code availability

The code for the bioinformatics analysis, alongside information on software and package versions, is available at https://github.com/stemcellbiologylab.

## Supplementary Information

## Acknowledgements

This work was supported by the DBT/Wellcome Trust India Alliance Fellowship (IA/I/15/2/502061) awarded to SK and intramural funds from Indian Institute of Science Education and Research Thiruvananthapuram (IISER TVM). IISER TVM Institutional animal facility is supported by funds from the Department of Science and Technology, Government of India (under FIST scheme; SR/FST/LS-II/2018/217). The authors wish to thank Prof. Catherine Verfaillie, KU Leuven, for providing material support and hosting S Khurana for performing experiments on animals. SH Mehatre and H Agrawal are supported by IISER TVM. IM Roy was supported by the senior research fellowship from University Grants Commission (UGC), India.

## Author Contributions

SH Mehatre: data curation, software, formal analysis, validation, investigation, visualization, and methodology.
H Agrawal: data curation, software, formal analysis, investigation, and visualization.
IM Roy: data curation, formal analysis, investigation, and methodology.
S Schouteden: investigation and methodology.
S Khurana: conceptualization, resources, supervision, funding acquisition, validation, visualization, project administration, and writing—original draft, review, and editing.

## Conflict of Interest Statement

The authors declare that they have no conflict of interest.

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
