## [Reviewer comments · Life Science Alliance]

Repeated proliferative events ameliorate age-associated accumulation of DNA damage in HSPCs

Shubham Mehatre, Harsh Agrawal, Irene Roy, Sarah Schouteden, and Satish Khurana
DOI: <https://doi.org/10.26508/lsa.202503337>

Corresponding author(s): Satish Khurana, Indian Institute of Science Education and Research Thiruvananthapuram

Review Timeline:

Submission Date:	2025-04-01
Editorial Decision:	2025-05-27
Revision Received:	2025-11-25
Editorial Decision:	2026-01-06
Revision Received:	2026-01-14
Accepted:	2026-01-20

Scientific Editor: Sarita Hebbar

Transaction Report:

May 26, 2025

Re: Life Science Alliance manuscript #LSA-2025-03337

Dr. Satish Khurana
Indian Institute of Science Education and Research Thiruvananthapuram
School of Biology
Maruthmala, Vithura
Thiruvananthapuram 695551
India

Dear Dr. Khurana,

Thank you for submitting your manuscript entitled "Proliferative events ameliorate DNA damage accumulation without affecting function in HSCs" to Life Science Alliance. The manuscript was assessed by two expert reviewers, whose comments are appended to this letter.

The reviewers noted that this study, on the impact of serial blood withdrawals on HSC proliferation in ageing, provides a significant advance to the field. That said they raised significant concerns that must be addressed before publication at LSA. Addressing these concerns might require new experiments and additions to the methods section, and we recommend that the following must be included:

1. Supplement the neutral comet assay with DNA damage/replication stress marker to account for oxidative damage and replication stress (Reviewer 2, comment 5)
2. A clear specification of sample size in terms of mice or single cells for every result. Additional samples must be analysed in case of low sample size, as indicated by Reviewer 2, in point 3
3. Definition of the specific cell types (HSC, HSPC, LSK etc) described in the manuscript, and use of the correct cell identity/nomenclature in a consistent fashion throughout the manuscript.
4. Details in the methods section for comet assay (Reviewer 2, point 3), for the experimental paradigm of serial bleeding and age of mice used (Reviewer 1, minor comments)
5. Removal of identical plots displayed as distinct results in different figures (Reviewer 2, point 1)

In line with the overall evaluation of the reviewers, we invite you to submit a revised manuscript addressing their comments.

When submitting the revision, please include a letter addressing the reviewers' comments point by point. While a rebuttal must respond to all points in some form, additional data to resolve these points (other than ones indicated above) is not required.

Thank you for this interesting contribution to Life Science Alliance. We are looking forward to receiving your revised manuscript.

Sincerely,

Sarita Hebbar, PhD
Scientific Editor

B. MANUSCRIPT ORGANIZATION AND FORMATTING:

Reviewer #1 (Comments to the Authors (Required)):

This manuscript addresses a topic with many unanswered questions: how HSC proliferation affects aging. Given the challenges involved, both technical and temporal-studies like this are especially valuable, and I commend the authors for their efforts. While in principle the observations have been reported before, there are discrepancies in the field and more reports are needed. Therefore, I recommend publication, pending revision. I do, however, have a few questions and concerns that I hope the authors will address.

Major Comments

- The manuscript states that "After confirming cell cycle entry in HSCs following repeated blood withdrawals (Fig S1A)," but the figure appears to show increased cycling in the LSK compartment rather than in HSCs specifically. Please clarify whether the data refer to HSPCs more broadly, or provide HSC-specific data. This distinction is particularly important given prior work from the Camargo lab suggesting that MPPs, not HSCs, bear the brunt of proliferative demand under stress.
- Similarly, the abstract refers to HSCs, while the measurements appear to include a broader HSPC population. Please ensure consistent terminology throughout, or clearly state when findings pertain to phenotypic HSCs versus broader progenitor compartments.

Minor Comments

- In the introduction, the distinction between LT-HSCs and phenotypic HSCs could be clarified. The sentence referencing Morrison et al. (1996) and Sudo et al. (2000) is somewhat confusing. Multiple studies have shown an accumulation of myeloid-biased HSCs with reduced competitive potential in aging, yet these cells remain multipotent. It would be helpful to rephrase this section for clarity.
- The authors cite Beerman et al. (2014) but not the follow-up study from 2024 (PMID: 39390312), which may address some discrepancies with other reports. Including this reference would strengthen the discussion.
- The schematic showing bleeding of older mice at different ages lacks clarity. Could the authors specify the age at which bleeding began, its frequency, and the interval between the last bleed and bone marrow analysis?
- The manuscript would benefit from more precise age terminology to help readers contextualize the findings. For instance, referring to 12-month-old mice as "early middle age" and 18-month-olds as "late middle age" may aid in interpretation.

- It is not clear whether live cell gating was used in the flow cytometry analysis. This should be stated explicitly in the methods and reflected in the gating strategy.
- Beerman et al. (PMID: 20304793) describe an age-related accumulation of CD150+high HSCs. Since the authors include CD150 in their panel, did they observe any proliferative stress-induced changes in CD150 expression?
- The transplantation model involved whole bone marrow rather than purified HSCs, and no secondary transplant was performed. While this is understandable, the authors should clarify that the observed effects may reflect changes in other progenitor populations as well, not just HSCs.
- The methods section or figure legend does not specify the number of biological replicates or the number of cells analyzed in the comet assay. This information is necessary to assess the robustness of the findings.

Editorial

- Some light editing for English clarity and grammar is recommended.
-

Reviewer #2 (Comments to the Authors (Required)):

The study investigates the effect of increased HSC proliferation induced by serial blood withdrawals during early adult life on aging of the hematopoietic system. While aged versus young control mice showed the expected age-associated changes in the cellular composition of the hematopoietic system and declining repopulation potential of HSC, inducing increased HSC proliferation had little to no effect on this age-associated phenotype. Moreover, the authors describe a slight reduction of accumulated DSBs in old mice that were subject to serial blood withdrawals and a clearance of DSB upon induction of proliferation, which was accompanied by transcriptional upregulation of DNA repair factors.

Against the general notion that repeated blood loss during life would drive aging of HSC the present study suggests no impact of serial bleeding on HSC functionality in aged mice. Even though merely descriptive, these findings are novel, of interest for the aging field and potentially clinically relevant. In the second part the authors investigate the effect of proliferation induced by bleeding on DNA damage and repair, corroborating previous research.

Specific comments:

1. Fig. 1F and Fig S1O appear to show identical plots. Also, recommend to order the plots in Fig. S1 accordingly to Fig. 1 and placing additional parameters at the end. This would make it easier to compare certain parameters over life time.
2. On page 7 the wrong figure (Fig. 2F) is indicated when the authors first refer to the comet assay experiment shown in Fig. 3A-C
3. From the neutral comet assay experiments in BM derived LSK cells (Fig. 3A-C) the authors conclude that the proportion of LSK cells with comet tail was elevated in the aged mice groups and remained unaffected by serial bleeding. Moreover, authors suggest a clearance of DNA damage following serial bleeding based on Olive tail moment analysis of single cells. How were cells with comet tail defined? Was a certain olive tail moment value set as a threshold? In any case a definition should be included in the Materials and Methods section. Also, why was '% of cells with comet tail' analyzed per mouse, but 'olive tail moment' per single cell? Overall this particular data set and especially the putatively reduced tail moment in response to the bleeding regime appears not very convincing, given that the number of analyzed samples is only 4, while in previous figures around 10 samples have been analyzed. Moreover, using single cell values will result in a p value <0.05 simply due to the large sample size. Thus, I recommend the authors to analyze additional samples or otherwise be more modest in their conclusions in this part. Moreover, they should indicate the number of samples (mice or single cells) analyzed in the figure legends.
4. More convincing is the clearance of DSB upon cell cycle entry in the short-term serial bleeding experiment. Here a difference in olive tail moment seems to be more pronounced compared to Fig. 3C. Indicating the fold changes in these plots or corresponding text would help to understand the magnitude of change.
5. As neutral comet assay is used for detecting DNA damage the analysis is restricted to DSBs. While their results suggest DSBs are cleared, others have described an increase of oxidative damage and replication stress after exit from quiescence (Walter et al, 2015; Flach et al, 2014). Therefore, it would be interesting to analyze additional DNA damage and replication stress marker. Otherwise authors should restrict their conclusions to DSBs and at least discuss the possibility of other damage types.

Authors' response to the reviewers' comments:

First of all, we thank the reviewers for going through the manuscript extremely carefully and critically. We appreciate the overall positive commentary from both of the reviewers. We feel satisfied that the message and importance of the presented study have been correctly conveyed. We have gone through each comment and suggestion made by the reviewers. We have found the comments very supportive, encouraging and insightful. We have made every effort to address the concerns raised and have added additional data/information wherever required. We believe that working on these suggestions has further confirmed our conclusions and enhanced the scope and impact of the manuscript. In the following section, we respond to each comment made by the reviewers and explain the changes that we have brought about in the manuscript to address the queries.

Referee #1:

This manuscript addresses a topic with many unanswered questions: how HSC proliferation affects aging. Given the challenges involved, both technical and temporal-studies like this are especially valuable, and I commend the authors for their efforts. While in principle the observations have been reported before, there are discrepancies in the field and more reports are needed. Therefore, I recommend publication, pending revision. I do, however, have a few questions and concerns that I hope the authors will address.

Authors: We thank the reviewer for the constructive and supportive feedback, we appreciate an overall positive evaluation of our work.

Major Comments

Comment 1. The manuscript states that "After confirming cell cycle entry in HSCs following repeated blood withdrawals (Fig S1A)," but the figure appears to show increased cycling in the LSK compartment rather than in HSCs specifically. Please clarify whether the data refer to HSPCs more broadly, or provide HSC-specific data. This distinction is particularly important given prior work from the Camargo lab suggesting that MPPs, not HSCs, bear the brunt of proliferative demand under stress.

Authors: We thank the reviewer for pointing out this important distinction. For the data presented in the Fig. S1A, we performed cell cycle analysis on Lin^c-Kit⁺Sca-1⁺ (LSK cells/HSPCs). In the results section of the submitted manuscript, these cells were incorrectly referred to as HSCs. We have changed this terminology in the revised manuscript to accurately reflect that the analysis pertains to the broader LSK/HSPC compartment. We agree with the reviewer that distinguishing between HSCs and downstream progenitors such as MPPs is important, as highlighted by previously published work.

Comment 2. Similarly, the abstract refers to HSCs, while the measurements appear to include a broader HSPC population. Please ensure consistent terminology throughout, or clearly state when findings pertain to phenotypic HSCs versus broader progenitor compartments.

Authors: We thank the reviewer for this comment and for pointing out this anomaly. We have revised the text to ensure consistent terminology throughout the manuscript. The revised text clearly distinguishes the phenotypic HSCs from a broader HSPC population, using the appropriate terms in each case based on the specific cellular markers analyzed.

Minor Comments

Comment 1. In the introduction, the distinction between LT-HSCs and phenotypic HSCs could be clarified. The sentence referencing Morrison et al. (1996) and Sudo et al. (2000) is somewhat confusing. Multiple studies have shown an accumulation of myeloid-biased HSCs with reduced competitive potential in aging, yet these cells remain multipotent. It would be helpful to rephrase this section for clarity.

Authors: We thank the reviewer for this suggestion. In the study from Morrison et al. (1996), long-term self-renewing multipotent progenitor cells were defined by the marker profile Thy-1^{lo}Sca-1^{hi}Lin⁻Mac-1⁻CD4⁻c-kit⁺, whereas Sudo et al. (2000) used CD34^{-/low}c-kit⁺Sca-1⁺Lin⁻ cells to investigate age-associated functional changes in HSCs. Given these differences in marker selection, we have revised the text to clarify the distinction between long-term HSCs (LT-HSCs) and phenotypic HSCs, and we now use distinct nomenclature to accurately reflect the definitions employed in each study. We have also rephrased the section to emphasize that, although aging leads to the accumulation of myeloid-biased HSCs with reduced competitive potential, these cells retain multipotency.

Comment 2. The authors cite Beerman et al. (2014) but not the follow-up study from 2024 (PMID: 39390312), which may address some discrepancies with other reports. Including this reference would strengthen the discussion.

Authors: This was in deed an oversight from our side to not have included the study pointed by the reviewer. We thank the reviewer for pointing this out. In the revised manuscript, we now discuss our results in light of these studies. Beerman et al. (2014) demonstrated that upon exit from quiescence ex vivo, aged HSCs activated DNA repair mechanisms and cleared pre-existing DNA damage. Contrarily, the report by Yanai H et al. (2024) showed that a sustained proliferative demand in vivo led to the accumulation of double-strand breaks. Consistent with our report however, these DNA damage accumulation was delinked from the functional changes in the stem cell population. The

revised manuscript discusses these results to better contextualize our findings and address potential discrepancies with previous reports.

Comment 3. The schematic showing bleeding of older mice at different ages lacks clarity. Could the authors specify the age at which bleeding began, its frequency, and the interval between the last bleed and bone marrow analysis?

Authors: We thank the reviewer for this helpful suggestion. We have revised the schematic and figure legends to clearly indicate the age at initiation, bleeding frequency, and the interval between the last bleed and analysis. In our experiments, repeated proliferative stress was induced in HSPCs by employing a long-term serial bleeding regimen. Mice aged 6 weeks were used for the protocol, and 200 μ l of blood was withdrawn weekly for 12 weeks via the tail clipping method. This was followed by fortnightly bleeding for up to 8 months of age, and monthly bleeding until the mice reached 12 months of age. Thereafter, mice were maintained without further intervention (except for blood withdrawal for analysis at 18 months of age) until sacrifice at 24 months. Peripheral blood analysis performed at 12, 18, and 24 months of age has been presented in the manuscript. The interval between the last bleed and bone marrow analysis was 6 months, as the mice were sacrificed at 24 months and the penultimate PB analysis was performed at 18 months of age.

Comment 4. The manuscript would benefit from more precise age terminology to help readers contextualize the findings. For instance, referring to 12-month-old mice as "early middle age" and 18-month-olds as "late middle age" may aid in interpretation.

Authors: We thank the reviewer for this constructive suggestion. We have revised the manuscript to include more precise age terminology, referring to 12-month-old mice as "early middle age" and 18-month-old mice as "late middle age," to aid in the interpretation and contextualization of the findings.

Comment 5. It is not clear whether live cell gating was used in the flow cytometry analysis. This should be stated explicitly in the methods and reflected in the gating strategy.

Authors: For flow cytometry based analysis, we performed immuno-staining on the mononuclear cells followed by fixation of the cells by 2% paraformaldehyde. Therefore, we did not have to apply live-cell gating. As the reviewer suggested, we have clarified this explicitly in the methods section, reflecting it in the revised gating strategy.

Comment 6. Beerman et al. (PMID: 20304793) describe an age-related accumulation of CD150+high HSCs. Since the authors include CD150 in their panel, did they observe any proliferative stress-induced changes in CD150 expression?

Authors: We thank the reviewer for this insightful comment. In the study by Beerman et al. (2010), a higher level of CD150 (SLAMF1) expression within the primitive HSC population identified as LSKCD34-Flt3⁻ cells that were also negative for CD48 expression. We used the marker system wherein CD150+CD48-LSK cells were identified as the primitive HSCs. To address the query from the reviewer, we have analyzed the expression of CD150 in CD48-LSK cells to examine the dynamics of CD150 expression with increasing age, and any effects following repeated proliferative events. We confirmed the previously reported increase in the expression of CD150 in HSPCs from aged mice. Importantly, we noted a further up-regulation of CD150 in the HSPCs from mice group that had undergone long-term blood withdrawal regimen (Fig. S2G). These results were intriguing for us as we did not see any difference between the two groups of aged mice despite a significant increase in CD150 expression in aged donor group. Hence, our results indicate that there might not be a significant link between increased CD150 expression and functional decline.

Comment 7. The transplantation model involved whole bone marrow rather than purified HSCs, and no secondary transplant was performed. While this is understandable, the authors should clarify that the observed effects may reflect changes in other progenitor populations as well, not just HSCs.

Authors: In deed, we used 16 weeks engraftment as the parameter for long-term repopulation potential of HSCs, and did not perform serial transplantation assays. Additionally, the results presented are based on whole BM transplantation and not sorted HSC. Therefore, we agree with the comment from the reviewer that our study reflects the function of hematopoietic system as a whole and not just the HSCs. This was intentionally designed to assess if there was any decline in the function at any level of the hematopoietic system. In the revised manuscript, we have re-emphasised this point for clarity.

Comment 8. The methods section or figure legend does not specify the number of biological replicates or the number of cells analyzed in the comet assay. This information is necessary to assess the robustness of the findings.

Authors: We thank the reviewer for raising this point. We noted the omission for some results and have revised the figure legends and methods section accordingly. We have explicitly stated the number of biological replicates and the number of single cells analyzed for each comet assay experiment. This will ensure the assessment of robustness of the findings presented in the manuscript. In fact, during revision of the manuscript, we have added additional replicates to increase robustness of the data presented.

Editorial Comment: Some light editing for English clarity and grammar is recommended.

Authors: We recognised some grammatical insufficiencies and typographical mistakes in the text. We apologise for these inadvertent mistakes. Careful editing has been done for the revised text. We thank the editor for this comment; we trust that working on the text has made it more clearly understandable.

Referee #2:

The study investigates the effect of increased HSC proliferation induced by serial blood withdrawals during early adult life on aging of the hematopoietic system. While aged versus young control mice showed the expected age-associated changes in the cellular composition of the hematopoietic system and declining repopulation potential of HSC, inducing increased HSC proliferation had little to no effect on this age-associated phenotype. Moreover, the authors describe a slight reduction of accumulated DSBs in old mice that were subject to serial blood withdrawals and a clearance of DSB upon induction of proliferation, which was accompanied by transcriptional upregulation of DNA repair factors.

Against the general notion that repeated blood loss during life would drive aging of HSC the present study suggests no impact of serial bleeding on HSC functionality in aged mice. Even though merely descriptive, these findings are novel, of interest for the aging field and potentially clinically relevant. In the second part the authors investigate the effect of proliferation induced by bleeding on DNA damage and repair, corroborating previous research.

Authors: We are grateful to the reviewer for going through the manuscript carefully, and for the positive comments on our study. We have gone through the comments made by the reviewer, to which we have responded through textual changes and additional data, wherever required. In the following section, we elaborate the changes made in the manuscript in response to these comments.

Comment 1. Fig. 1F and Fig S1O appear to show identical plots. Also, recommend to order the plots in Fig. S1 accordingly to Fig. 1 and placing additional parameters at the end. This would make it easier to compare certain parameters over life time.

Authors: We recognize this oversight and thank the reviewer immensely for pointing this out. This was an inadvertent mistake that we have rectified in the revised manuscript. Additionally, the comment on reordering the the panels was useful, and we have made suggested changes in the revised manuscript.

Comment 2. On page 7 the wrong figure (Fig. 2F) is indicated when the authors first refer to the comet assay experiment shown in Fig. 3A-C.

Authors: This was an inadvertent mistake and has been corrected in the revised manuscript.

Comment 3. From the neutral comet assay experiments in BM derived LSK cells (Fig. 3A-C) the authors conclude that the proportion of LSK cells with comet tail was elevated in the aged mice groups and remained unaffected by serial bleeding. Moreover, authors suggest a clearance of DNA damage following serial bleeding based on Olive tail moment analysis of single cells. How were cells with comet tail defined? Was a certain olive tail moment value set as a threshold? In any case a definition should be included in the Materials and Methods section. Also, why was '% of cells with comet tail' analyzed per mouse, but 'olive tail moment' per single cell? Overall this particular data set and especially the putatively reduced tail moment in response to the bleeding regime appears not very convincing, given that the number of analyzed samples is only 4, while in previous figures around 10 samples have been analyzed. Moreover, using single cell values will result in a p value <0.05 simply due to the large sample size. Thus, I recommend the authors to analyze additional samples or otherwise be more modest in their conclusions in this part. Moreover, they should indicate the number of samples (mice or single cells) analyzed in the figure legends.

Authors: We thank the reviewer for this comment. In the revised manuscript we have provided details of the methods employed to perform the comet assays and for the image analysis. The methods that we have followed have been widely published for a variety of cell types (She W e al. in *Blood* 2017, Beerman I et al. in *Cell Stem Cell* 2014); broadly, we have followed standard practices employed in these experiments. We considered the cells with >5% DNA in comet tail as positive for DNA damage and compared different samples. The methods section has been revised to include the details of analysis pipeline used.

In Fig. 3B, we have shown if there was any change in the proportion of cells with DNA damage above the threshold, following the long-term bleeding regimen. This analysis was performed on the population of HSPCs, unlike Fig. 3C wherein DNA damage per cell is shown. Olive tail moment takes into account the proportion of DNA in the tail and the length of tail at single cell level. We believe that the text was not clearly reflecting this distinction leading to this query. We have modified the text and included the details for more clarity.

Regarding the reviewer's comment on statistical analysis, we acknowledge that using single-cell values will involve large number of measurements. In order to address this comment, we have added more data and presented comparison between the samples taking into consideration mean values for each independent experiment. The average Olive tail moment for young controls, aged controls and aged donor groups was 5.206 ± 1.034 , 14.20 ± 2.236 and 7.341 ± 0.9363 , respectively. Regardless of the analytical approach used, the aged donor mice that underwent long-term bleeding

regimen during their lifetime, showed significant clearance of DSBs. The number of biological and technical replicates have been added in the figure legends as per the reviewers suggestion.

Comment 4. More convincing is the clearance of DSB upon cell cycle entry in the short-term serial bleeding experiment. Here a difference in olive tail moment seems to be more pronounced compared to Fig. 3C. Indicating the fold changes in these plots or corresponding text would help to understand the magnitude of change.

Authors: We appreciate the reviewer's assessment and agree that it was perhaps the most significant finding of our study. In deed there was a significant clearance (Olive tail moment of 17.55 ± 0.4664 and 5.634 ± 0.2085 for aged controls and aged bled mice, respectively). DSBs following induction of proliferative events in aged HSCs. While we followed standard practices to show data from comet assays, following the reviewers suggestion, we have added absolute values of Olive tail moment in linear scale for better evaluation of fold change. As per the reviewer's suggestion, we have made changes in the text to reflect the fold change in the values.

Comment 5. As neutral comet assay is used for detecting DNA damage the analysis is restricted to DSBs. While their results suggest DSBs are cleared, others have described an increase of oxidative damage and replication stress after exit from quiescence (Walter et al, 2015; Flach et al, 2014). Therefore, it would be interesting to analyze additional DNA damage and replication stress marker. Otherwise authors should restrict their conclusions to DSBs and at least discuss the possibility of other damage types.

Authors: We completely agree with this assertion, and thank the reviewer for this important comment. To address this concern, we have now performed experiments to assess the activation of DNA damage repair (DDR) pathways and replication stress markers. For this analysis, we used short-term bleeding regimen in aged mice and performed γ H2AX staining on LSK cells. Appearance of γ H2AX foci is indicative of activation of DNA damage repair pathways in response to the occurrence of DSBs (Bassing CH et al. in *PNAS* 2002, Celeste A et al. in *Cell* 2003, Downs JA et al. in *Mol. Cell* 2004). While the appearance of γ H2AX foci could be used to assess the efficiency of DSB repair efficiency, it has also been used to quantify the appearance of DSBs. Therefore, we decided to compare the foci that appeared in the LSK cells from aged mice that underwent short-term bleeding regimen (aged bled) with aged control animals. Our results showed higher level of γ H2AX foci in the HSPCs from 'aged bled' mice group. Seen together with the comet assay data that showed robust clearance of DSBs following activation of proliferation in aged HSPCs, these results point towards rapid DDR activation.

To further expand the analysis beyond DSBs, we performed PARP1 staining, which can reflect activation of the DNA damage response in the context of oxidative DNA damage response (Hsu et al. *Leukemia*, 2018). Consistent with our γ H2AX data, PARP1 staining was also increased upon induction of proliferation, supporting the conclusion that serial bleeding activates DNA damage response signaling. These findings are in agreement with our analysis on scRNA seq data and subsequently quantitative RT-PCR based confirmation of gene expression changes upon cell cycle entry. Put together, these results have demonstrated that blood withdrawal induced proliferative events in aged HSPCs can activate DDR pathways leading to clearance of DNA damages accumulated over the lifetime. Our results not only corroborate earlier findings but also provide in vivo relevance and mechanistic insights. These findings also question the link between DNA damage accumulation and functional decline in HSPCs during aging process.

January 6, 2026

RE: Life Science Alliance Manuscript #LSA-2025-03337R

Dr. Satish Khurana
Indian Institute of Science Education and Research Thiruvananthapuram
School of Biology
Maruthmala, Vithura
Thiruvananthapuram 695551
India

Dear Dr. Khurana,

Thank you for submitting your manuscript, "Repeated proliferative events ameliorate age-associated accumulation of DNA damage in HSPCs". Thank you for your patience awaiting our decision, which was delayed due to editor availability and previous delays in securing reviewer comments.

Your revised manuscript was evaluated by one of the original reviewers whose comments are appended below. As you will read, the reviewer is satisfied that the revised work has addressed their concerns.

In line with the reviewers' evaluation, we would be happy to publish your paper in Life Science Alliance pending final revisions necessary to meet our formatting guidelines.

- The title, abstract, and text appear to use HSC and HSPC interchangeably. Please review the entire text, including the title and abstract, to ensure consistent usage and invoke progenitor cells (HSPC) only where this term is warranted. Please also ensure to spell out each in their first use (i.e. in the first sentence of the abstract).
- Please define "*" in the legends for Figure 1, Figure 3.
- Please provide a scale bar for images in Figure 4 along with size information in the legend.
- Thank you for providing a schematic of your workflow. Please refer to the schematic in the relevant methods section.
- In the methods section please provide concentration details for antibodies if available, and please state method for fold-change calculation in qRT-PCR assays. Kindly refer to Table 1 in the description under "Immunostaining and imaging".
- LSA does not permit citation of "data not shown," "manuscript in preparation," "manuscript submitted," etc., in any section of the manuscript. Please remove "data not shown" and the associated claim or include the evidence that is referenced. Please refer to page 7, "In our experiments, we did not observe any significant change in the multi-lineage donor derived engraftment between the three groups of mice up to a period of 12 weeks (data not shown)."
- Please upload all figure files as individual ones, including the supplementary figure files; all figure legends should only appear in the main manuscript file.
- It is recommended to exclude figures from the manuscript text and upload them separately.
- Please add the X and Bluesky handles of your host institute/organization, as well as your own and/or one of the authors, in our system.
- Please add your main, supplementary figure, and table legends to the main manuscript text after the references section.
- Please be sure that the authorship listing and order are correct.

A. FINAL FILES:

B. MANUSCRIPT ORGANIZATION AND FORMATTING:

Thank you for your attention to these final processing requirements. Please revise and format the manuscript and upload materials as soon as you are able.

Sincerely,

Sarita Hebbar, PhD
Scientific Editor
Life Science Alliance
<http://www.lsajournal.org>

Reviewer #2 (Comments to the Authors (Required)):

The authors have greatly improved the manuscript, providing additional data and explanations in the text. Also, the point by point responses to the reviewer comments are adequate.
Thus, I recommend this manuscript for publication in LSA

January 20, 2026

RE: Life Science Alliance Manuscript #LSA-2025-03337RR

Dr. Satish Khurana
Indian Institute of Science Education and Research Thiruvananthapuram
School of Biology
Maruthmala, Vithura
Thiruvananthapuram 695551
India

Dear Dr. Khurana,

Thank you for submitting your revised manuscript entitled "Repeated proliferative events ameliorate age-associated accumulation of DNA damage in HSPCs". It is a pleasure to let you know that your manuscript is now accepted for publication in Life Science Alliance. Congratulations on this interesting work.

DISTRIBUTION OF MATERIALS:

Again, congratulations on a very nice paper. I hope you found the review process to be constructive and are pleased with how the manuscript was handled editorially. We look forward to future exciting submissions from your lab.

Sincerely,

Sarita Hebbar, PhD
Scientific Editor
Life Science Alliance
<http://www.lsajournal.org>